# Transient Hepatitis B Surface Antigenemia Following Immunization with the Adjuvanted Hepatitis B Vaccine Fendrix^®^

**DOI:** 10.3390/vaccines13121216

**Published:** 2025-11-30

**Authors:** Virginia Fernández-Espinilla, Paula Ardura-Agudín, Daniel Leonardo Sánchez-Carmona, Sandra Sanz-Ballesteros, Kenia Piedad Cobo-Campuzano, Cristina Hernán-García, José Javier Castrodeza-Sanz, María del Camino Prada-García

**Affiliations:** 1Department of Preventive Medicine and Public Health, University of Valladolid, 47005 Valladolid, Spain; 2Preventive Medicine and Public Health Service, Hospital Clínico Universitario de Valladolid, 47003 Valladolid, Spain; 3Nephrology Service, Hospital Clínico Universitario de Valladolid, 47003 Valladolid, Spain; 4Dermatology Service, Complejo Asistencial Universitario de León, 24008 León, Spain

**Keywords:** chronic kidney disease, hepatitis B surface antigen, vaccination, hepatitis B virus

## Abstract

Background: Serological screening for HBV is standard in hemodialysis, and vaccination is recommended for non-immune patients. Objective: To determine the cause of positive HBsAg detected shortly after vaccination. Methods: We conducted a retrospective study in a tertiary hemodialysis unit. Patients with HBsAg reactivity after receiving the adjuvanted HBV vaccine (Fendrix^®^) were followed with serial serology until HBsAg clearance. Results: Forty-four patients were monitored; seven (15.9%) tested HBsAg-positive 1–7 days post-vaccination, with no evidence of acute hepatitis, prior HBV infection, transplantation, or chronic immunosuppression. Six cleared HBsAg on repeat testing; one remained positive until day 19, with HBsAg as the only marker. Conclusions: Vaccine-related transient HBsAg antigenemia can occur shortly after immunization. Recognizing this phenomenon and timing routine serology appropriately can prevent misinterpretation and unnecessary workups in CKD patients.

## 1. Introduction

The World Health Organization (WHO) estimated that 254 million individuals were living with chronic hepatitis B virus (HBV) infection in 2022, with approximately 1.2 million new cases occurring each year [1].

The epidemiological data presented in this section are based on the most recent reports available at the time of manuscript preparation and are therefore not restricted to the calendar year of the clinical data collection (May–June 2023).

Spain remains classified as a low-prevalence country for hepatitis B, consistently reporting an annual incidence below 2 cases per 100,000 population. In 2023, 359 cases were recorded, corresponding to an incidence rate of 0.50 per 100,000. The majority of new infections are identified in young adult immigrants, while comprehensive prevention strategies have led to the successful elimination of mother-to-child (vertical) transmission. The estimated prevalence of chronic HBV carriers in the general population ranges from 0.2% to 0.5%. Surveillance and monitoring of hepatitis B are conducted through the National Epidemiological Surveillance Network [2].

Patients with chronic kidney disease (CKD) are especially susceptible to infections preventable by vaccination due to their compromised immune function and frequent exposure to parenteral interventions, such as hemodialysis. Additionally, factors such as advanced age, diabetes mellitus, cardiovascular comorbidities, and a chronic pro-inflammatory state further increase their risk of infection-related complications [3,4].

Patients with advanced CKD have reduced vaccine immunogenicity and faster antibody waning, underscoring the need for tailored HBV vaccination strategies [5,6]. In chronic hemodialysis settings, repeated exposure to blood and body fluids makes HBV a particular concern; accordingly, KDIGO recommends dedicated infection-control precautions for HBV-positive patients in dialysis units [3].

Due to the diminished immunogenicity frequently observed in pre-dialysis and dialysis patients with CKD, compared to immunocompetent individuals, alternative immunization strategies may be necessary. These may include higher antigen doses, the use of adjuvanted vaccine formulations, or more frequent booster administrations to achieve adequate protection [7].

Currently, all hemodialysis units recommend serological screening and verification of HBV immunization, with vaccination indicated for any non-immunized patient to prevent infection and minimize the consequent risks of chronicity, cirrhosis, and hepatocellular carcinoma [5]. HBV screening is performed using HBsAg, Anti-HBc, and Anti-HBs markers in quarterly blood tests [8].

For CKD patients aged 15 years or older, the adjuvanted HBV vaccine (Fendrix^®^), composed of 20 µg of HBsAg with the AS04C adjuvant system, is recommended to enhance humoral and cellular immunity. The adjuvant contains 50 µg of 3-O-desacyl-4′-monophosphoryl lipid A (MPL) in 500 µg of aluminum phosphate. The vaccine antigen is produced by recombinant DNA technology in Saccharomyces cerevisiae [9]. The vaccination schedule consists of 4 doses at months 0, 1, 2, and 6 [10].

Post-vaccination monitoring should be carried out 1–2 months after the last dose and annually for all patients undergoing chronic hemodialysis. Seroconversion or response is defined by Anti-HBs titers > 10 IU/L; if not achieved, a booster dose is recommended [9].

A patient is considered a non-responder if titers remain below 10 IU/L after the second vaccination cycle. Some authors have suggested defining a complete protective response with Anti-HBs levels > 100 IU/L [11,12]. Despite the fact that 15–50% of vaccinated individuals may lose detectable Anti-HBs over time, they remain protected due to the cellular immune memory induced by vaccination [13]. This report provides the first clinical series describing vaccine-related HBsAg antigenemia following Fendrix^®^ in CKD patients on hemodialysis and highlights practical considerations for the timing of HBsAg testing in this population.

## 2. Materials and Methods

A descriptive observational retrospective study was conducted in the Hemodialysis Unit of the Nephrology Department at the Hospital Clínico Universitario de Valladolid, which provides care to 44 patients undergoing renal replacement therapy. The study was carried out during May and June 2023 with the aim of documenting the unexpected detection of hepatitis B surface antigen (HBsAg) reactivity in a group of patients recently vaccinated with Fendrix. The investigation was performed in collaboration with the Department of Preventive Medicine and Public Health at the same institution.

Population: The study included all 44 chronic hemodialysis patients monitored in the unit. As per standard protocol, patients underwent quarterly serological testing for viral infection markers, including HBsAg.

The primary vaccination cycle was defined as the administration of four doses of the Fendrix^®^ vaccine at months 0, 1, 2, and 6. A booster dose was administered as a single additional dose to patients who had previously completed the primary vaccination cycle.

Event Detection: Initial HBsAg screening was performed using a chemiluminescent microparticle immunoassay (CMIA; Abbott Alinity^®^ HBsAg Qualitative II, Abbott Park, IL, USA). All initially reactive results were re-tested from a fresh aliquot and confirmed with the manufacturer-recommended neutralization (anti-HBs blocking) procedure [14].

In addition to initial HBsAg screening, all patients underwent confirmatory serological and molecular testing, including anti-HBs quantification, total anti-HBc, HBV DNA quantification by PCR, and serum alanine aminotransferase (ALT) measurement.

Infection-control response: Per institutional policy, any HBsAg-reactive result triggers an epidemiologic surveillance protocol (case review, repeat serology, confirmatory tests, and unit-level infection-control assessment).

Laboratory Testing: All seven patients underwent confirmatory testing to rule out active HBV infection. These included anti-HBs testing, HBV DNA quantification, anti-HBc serology, and alanine aminotransferase levels (ALT).

Collected Variables: Retrospective data were extracted from electronic medical records and included age, sex, immunosuppressive comorbidities, duration of hemodialysis, hepatitis B vaccination history (date, vaccine type, schedule, batch number, and route of administration), and serological test results before and after vaccination.

Time to HBsAg clearance was calculated as the days between the index HBsAg-reactive sample and the first subsequent non-reactive sample. When no intermediate tests were available, clearance time was treated as interval-censored and reported as “≤X days,” where X is the length of the testing interval. Patients who did not achieve a non-reactive result during follow-up were right-censored at the date of their last available HBsAg measurement.

Ethics: This study was approved by the Institutional Review Board (PI-24-664-C). Given the retrospective design, informed consent was not required.

## 3. Results

Between May and June 2023, the Hemodialysis Unit provided care to 44 patients enrolled in the Chronic Hemodialysis Program, 68% (*n* = 30) of whom were male. Male age: mean 71.97 years (SD 14.75); female age: mean 69.07 years (SD 12.65).

Seven patients had a prior history of kidney transplantation and were receiving ongoing immunosuppressive therapy with either tacrolimus or everolimus. Two patients had documented past infections with the hepatitis B virus.

During the observation period, two patients died—one due to complications associated with CKD and another due to pancreatic cancer. Additionally, one patient was discharged from the unit following successful kidney transplantation.

Routine serological follow-ups detected 7 patients (15.9%) with positive HBsAg. None of them exhibited symptoms or clinical signs suggestive of acute hepatitis, nor did they have a history of past HBV infection. In addition to CKD, three patients had monoclonal gammopathies that were being monitored by the Hematology department with no active treatment (“*watch-and-wait*” approach). None of the patients with transient antigenemia had undergone transplantation or received long-term immunosuppressive therapy.

These serological findings were reviewed in collaboration with the Preventive Medicine Department to exclude active hepatitis B infection. HBsAg testing was performed both before and after vaccination as part of routine quarterly viral surveillance in the hemodialysis unit. Serial serologic testing was performed, and all patients were negative for both anti-HBs and anti-HBc prior to vaccination. Post-vaccination evaluations included anti-HBc, HBV DNA (via PCR), and alanine aminotransferase (ALT) levels, all of which were within normal limits (no elevations detected).

A common feature among patients with positive HBsAg was recent vaccination: all seven had received a dose 1–7 days prior to testing (mean 3.85 days; SD 2.74); two were undergoing a primary series and five had received a booster. In the full cohort (n = 44), only these seven had been vaccinated within the preceding 20 days; none of the remaining 37 had recent vaccination, and none showed HBsAg reactivity.

No adverse events or dialysis equipment failures were reported. Additionally, it was confirmed that these patients had not shared dialysis machines with the two patients who had a history of HBV infection.

On repeat testing, six patients had cleared HBsAg, while one patient remained HBsAg-positive until 19 days post-vaccination. Given that time to HBsAg clearance was only known to occur on or before the first non-reactive test, we summarized it using the upper bounds: median ≤ 4 days (IQR ≤2–≤7). Landmark proportions were ≤1 day: 1/7 (14.3%), ≤3 days: 3/7 (42.9%), ≤7 days: 6/7 (85.7%), and ≤19 days: 7/7 (100%). None of the patients achieved a seroprotective anti-HBs response (<10 IU/L).

The characteristics of patients with transient vaccine-related HBV antigenemia are summarized below in Table 1.

## 4. Discussion

HBV is a DNA virus transmitted via percutaneous, sexual, and perinatal exposure, affecting a substantial number of individuals globally. Hepatitis B surface antigen (HBsAg) detection is central to diagnosing and managing HBV infection. However, HBsAg reactivity shortly after vaccination may reflect transient, vaccine-derived antigenemia, which can complicate interpretation and lead to unnecessary interventions [15]. In recently vaccinated patients, recombinant HBsAg contained in current vaccine formulations can be detected by highly sensitive and specific assays without indicating infection; this yeast-expressed subunit antigen is non-infectious and poses no risk of vaccine-transmitted HBV [16].

A plausible mechanism is that, following intramuscular or subcutaneous administration of the aluminum-adjuvanted vaccine, most of the HBsAg remains extracellular at the injection site, with only a small proportion passively entering the circulation and thus being detectable by CMIA [14]. In addition, the Abbott Alinity^®^ HBsAg CMIA is a highly sensitive assay, capable of detecting very low concentrations of circulating HBsAg, which may contribute to the identification of transient, vaccine-derived antigenemia in the absence of true infection. The balance between macrophage/antigen-presenting cell internalization of alum-adsorbed HBsAg and its persistence in the extracellular space with subsequent bloodstream filtration remains insufficiently understood and warrants further study [17].

Although additional biochemical or immunological analyses would provide valuable mechanistic insight, such experiments were not feasible in the context of this retrospective study. No biological samples were stored, and the dataset was derived entirely from routine clinical surveillance rather than from a prospective protocol designed to measure immunological markers or antigen kinetics. As a result, direct mechanistic confirmation—such as quantifying circulating antigen or evaluating immune activation markers—was beyond the scope of the present work.

However, it should be noted that the Abbott Alinity^®^ HBsAg assay used in this study has a very high analytical sensitivity (0.017 IU/mL) and a specificity > 99%, making it capable of detecting even very low levels of circulating, vaccine-derived antigen.

Several mechanistic hypotheses may nevertheless account for this phenomenon, supported by prior experimental research. These include extracellular persistence of alum-adsorbed antigen and slow-release kinetics from the injection site. Experimental studies by Iyer et al. and Bauer et al. [18,19] provide biological plausibility for these mechanisms, although confirmation in our cohort was not possible due to the retrospective design and lack of stored biological samples.

Pre-existing immunity from past vaccination may accelerate antigen clearance through a rapid anamnestic response, thereby shortening the duration of transient antigenemia. However, prior immunity does not explain the presence of vaccine-related HBsAg in circulation. In our series, more than half of the patients with transient antigenemia had previously been vaccinated, which is consistent with the known waning of detectable anti-HBs over time in CKD patients. In this context, anti-HBs measured concurrently with HBsAg was below the assay threshold at the time of antigen detection, so contemporaneous neutralization could not be demonstrated; nevertheless, the observed clearance pattern is compatible with an anamnestic response and accelerated antigen clearance.

Detectability may vary across assays; however, this yeast-expressed subunit antigen is non-infectious, and transient vaccine-related antigenemia has been reported with multiple hepatitis B vaccines (Table 2).

The literature describes cases of transient HBsAg positivity in patients vaccinated against HBV with a non-adjuvanted recombinant vaccine. In 1993 and 1994, Challapalli et al., Bernstein et al., and Weintraub et al. reported detection of HBsAg in newborns vaccinated with Engerix B^®^ [19,20,21]. Similarly, Kloster et al. (1995) identified transient vaccine-related HBsAg antigenemia in nine blood donors vaccinated with Engerix B^®^ 1 to 3 days before donation [22]. In 1996, Janzen et al. described transient HBsAg positivity in hemodialysis patients vaccinated with Engerix B^®^ [23]. Subsequently, in 1998, Olde and Garcia reported a hemodialysis patient vaccinated with Engerix B^®^ who tested positive for HBsAg; this prompted the center to conduct serological tests on seven other patients, revealing a 50% incidence of transient vaccine-related antigenemia [24].

Rysgaard et al. (2012) further demonstrated that hepatitis B vaccination can induce transient HBsAg positivity in hemodialysis patients, typically resolving within 14 days [29]. In a study of immunocompetent patients (n = 117), Anjum (2014) reported a 2.5% incidence of transient vaccine-related HBsAg antigenemia results following Engerix B^®^ vaccination [30]. Transient vaccine-related HBsAg antigenemia has also been documented with other vaccines, including the adjuvanted Heplisav-B^®^ [32]. Although both Fendrix^®^ and Heplisav-B^®^ are adjuvanted formulations, they differ in their adjuvant systems (AS04C vs. CpG 1018), which may influence their immunostimulatory profiles and antigen kinetics. Similar observations have also been described with combination hepatitis B and hepatitis A vaccines such as Twinrix^®^ [27,31].

Similar findings have also been reported with other adjuvanted hepatitis B vaccines, such as Heplisav-B^®^, which contains the CpG 1018 adjuvant. In contrast, Fendrix^®^ incorporates the AS04C adjuvant system, combining monophosphoryl lipid A with aluminum phosphate. Despite these differences in adjuvant composition and immunostimulatory pathways, both vaccines have been associated with transient HBsAg detection shortly after administration.

The experimental studies by Ziaee et al. (2010) and Otağ (2003), among the few conducted with this design, demonstrated that vaccination with different recombinant hepatitis B vaccines can induce transient vaccine-related HBsAg antigenemia in healthy adults [26,28]. Notably, this effect was short-lived, resolving within 2 to 4 days in the study by Ziaee et al. and within 3 days or less in the study by Otağ. These findings highlight the importance of considering this temporary response when interpreting serological test results shortly after vaccination [26,28].

Active HBV infection was ruled out in all patients. Notably, none of the patients developed a protective Anti-HBs titer during the one-month follow-up. This lack of antibody response, likely related to their impaired immunity (e.g., due to CKD, advanced age, and coexisting hematologic conditions in some patients), may help explain why vaccine-derived HBsAg persisted in some patients for days. However, given the small sample, we cannot definitively link immunosuppression status to prolonged antigenemia.

The absence of measurable anti-HBs in all cases is consistent with the well-documented impaired vaccine responsiveness in CKD patients, particularly in older individuals and in those with chronic inflammation or hematologic comorbidities, which are highly prevalent in this cohort. These immunogenicity deficits may contribute to delayed or insufficient neutralization of circulating antigen in the early post-vaccination period.

This study presents the first scientific evidence of transient vaccine-related HBsAg antigenemia associated with the adjuvanted recombinant hepatitis B vaccine Fendrix^®^ in patients with CKD undergoing hemodialysis. With a median antigenemia clearance time of ≤4 days (IQR ≤2–≤7), this highlights a practical diagnostic challenge in a vulnerable population and the need to consider recent vaccination when interpreting HBsAg results.

Previous studies in chronically infected patients (not limited to those on hemodialysis) have described urinary detection of HBV components, including HBsAg and HBV DNA, suggesting that the clearance kinetics of viral markers may differ from those observed after vaccination. Although this phenomenon is unrelated to vaccine administration, it provides additional context for understanding the transient appearance of circulating HBsAg. This clarification does not modify our interpretation that the antigenemia observed in our cohort is vaccine-related and self-limited.

Our series focuses on vaccine-related, short-lived serum HBsAg antigenemia, with undetectable serum HBV DNA and no urine testing performed; these observations are relevant for infection-control considerations but do not alter our interpretation or recommendations regarding the timing of post-vaccination serological testing.

This study is inherently limited by its small sample size, as only seven cases of transient HBsAg antigenemia were identified within a total cohort of 44 hemodialysis patients and a retrospective design. Clearance times were interval-censored by the spacing of serologic tests. While guidelines recommend annual HBV serology in CKD, our unit traditionally performs quarterly testing irrespective of recent vaccination. Based on these findings, we now defer routine serology for at least one week post-immunization and interpret HBsAg with caution for up to four weeks to avoid unnecessary investigations and anxiety.

Because the phenomenon is rare, unpredictable, and detectable only when serology coincides with very recent vaccination, no additional cases can be retrieved retrospectively, and expanding the sample prospectively within the same unit is not feasible. Nevertheless, we deliberately included the entire population at risk during the study period, which preserves internal validity despite the small number of events. The small number of cases did not allow for multivariable analyses to explore potential predictors of antigenemia or clearance kinetics, and therefore the analysis was limited to descriptive statistics.

## 5. Conclusions

In this study of chronic hemodialysis patients vaccinated with the adjuvanted hepatitis B vaccine Fendrix^®^, transient hepatitis B surface antigenemia was observed in 15.9% of the cohort, occurring within 1–7 days after immunization and clearing by ≤19 days in all cases. None of the affected patients showed clinical, biochemical, or virological evidence of acute hepatitis B, and HBV DNA remained undetectable, supporting a vaccine-related, non-infectious origin of HBsAg positivity.

These findings provide the first evidence of transient HBsAg antigenemia associated with Fendrix^®^ in patients with chronic kidney disease on hemodialysis and highlight an important and previously underrecognized diagnostic pitfall in routine surveillance. In settings where HBsAg is monitored regularly, recent vaccination should be systematically considered when interpreting reactive results to avoid unnecessary alarm, additional testing, and infection-control interventions. Based on our experience, we recommend deferring routine HBsAg testing for at least one week after hepatitis B vaccination and interpreting positive results with caution for up to four weeks in hemodialysis units.

## Figures and Tables

**Table 1 vaccines-13-01216-t001:** Characteristics of patients with transient vaccine-related HBsAg antigenemia following administration of the adjuvanted hepatitis B vaccine Fendrix^®^. The table includes demographic characteristics, vaccination schedule, pre- and post-vaccination serology, and time to HBsAg clearance. Abbreviations: M = male; F = female; HBsAg = hepatitis B surface antigen; anti-HBs = hepatitis B surface antibody; anti-HBc = hepatitis B core antibody; ALT = alanine aminotransferase; HBV DNA = hepatitis B virus DNA; S/CO = signal-to-cutoff ratio.

ID	Sex	Age (Years)	Pre-Vaccination–Test Interval Mean 35.29 Days (SD = 14.39)			Vaccination		Post-Vaccination–Test Interval Mean 3.85 Days (SD = 2.74)					Time to Clearance (Days) * Median ≤ 4 Days (IQR ≤ 2–≤ 7 Days)	Other Causes of Immunosuppression
			HBsAg (S/CO)	Anti-HBs (IU/L)	Anti-HBc	Date	Schedule	HBsAg (S/CO)	Anti-HBs (IU/L)	Anti-HBc	HBV DNA	Alanine Aminotransferase Levels (ALT) IU/L		
1	M	79	Non-reactive (0.34)	Negative (<10)	Negative	2/5/23	Booster	Reactive (2.37)	Negative (<10)	Negative	Not detected	4	≤3	IgA monoclonal gammopathy
2	M	54	Non-reactive (0.29)	Negative (<10)	Negative	10/5/23	Primary Cycle	Reactive (3.11)	Negative (<10)	Negative	Not detected	7	≤1	None
3	M	57	Non-reactive (0.29)	Negative (<10)	Negative	8/5/23	Primary Cycle	Reactive (4.62)	Negative (<10)	Negative	Not detected	3	≤19	None
4	F	76	Non-reactive (0.31)	Negative (<10)	Negative	23/5/23	Booster	Reactive (4.69)	Negative (<10)	Negative	Not detected	8	≤7	None
5	M	80	Non-reactive (0.28)	Negative (<10)	Negative	30/6/23	Booster	Reactive (3.06)	Negative (<10)	Negative	Not detected	13	≤7	Primary amyloidosis
6	M	68	Non-reactive (0.31)	Negative (<10)	Negative	24/5/23	Booster	Reactive (2.36)	Negative (<10)	Negative	Not detected	9	≤2	Monoclonal gammopathy of uncertain significance
7	M	80	Non-reactive (0.38)	Negative (<10)	Negative	24/5/23	Booster	Reactive (1.37)	Negative (<10)	Negative	Not detected	10	≤4	None

* Times are interval-censored; “≤X days” indicates that HBsAg clearance occurred on or before day X. Median and IQR represent the upper bounds only. S/CO is unitless; results are considered reactive if ≥1.0.

**Table 2 vaccines-13-01216-t002:** Summary of transient vaccine-related HBV antigenemia studies.

Author	Year	Design	Vaccine	Age	Population	Positive/Total	Duration of HBsAg Clearance
Challapali et al. [19]	1993	Prospective	Engerix-B^®^	Newborns	Newborns	10/18 55.6%	7–17 days
Bernstein et al. [20]	1994	Prospective	Engerix-B^®^	Newborns	Newborns > 2000 g NICU	19/25 76.0%	2–8 days
Weintraub et al. [21]	1994	Prospective	Engerix-B^®^	Newborns	Newborns	8/47 17.0%	16–17 days
Kloster et al. [22]	1995	Prospective	Engerix-B^®^	NA	Blood donors	8/19 42.1%	3 days
Janzen et al. [23]	1996	Case Series	Engerix-B^®^	NA	CKD	6	20 days
Olde and Garcia [24]	1998	Case Series	Engerix-B^®^	NA	CKD	8	<14 days
Ly et al. [25]	2002	Prospective	Engerix-B^®^, Recombivax^®^	Mean: 56 ± 18.8 years	CKD	9/2400 4%	1–28 days
Otag [26]	2003	Experimental	Engerix-B^®^, Hepavax Gene^®^, Gen Hevac B^®^	18–60 years	Healthy Adults	3/44 6.8%	3 days
De Schryver et al. [27]	2004	Retrospective	Twinrix^®^	Range: 33–37 years	Healthy adults	4	1 day
Ziaee et al. [28]	2010	Experimental	Engerix-B^®^, Hepavax-Gene^®^	20–22 years	Healthy medicine students	9/62 14.5%	
Rysgaard et al. [29]	2012	Retrospective	Engerix-B^®^, Twinrix^®^	>18 years	CKD in 10/11	11/34 32.4%	14 days
Anjum et al. [30]	2014	Retrospective	Engerix-B^®^	Mean: 31.34 years	Healthy adults	3/117 2.6%	0 days
Lee et al. [31]	2014	Case Report	Twinrix^®^	51 years	Adult		
Corsini et al. [32]	2020	Retrospective	Heplisav-B^®^	Median: 48 years	CKD in 5/6 positive results	6/39 15.4%	Mean 17 days
This study	2025	Retrospective	Fendrix^®^	Mean 70 years (SD = 11.21)	CKD	7/44 15.9%	≤1–≤19 days (median ≤ 4)

## Data Availability

The raw data supporting the conclusions of this article (Appendix A) will be made available by the authors upon request.

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
