# Peer review of "Transient Hepatitis B Surface Antigenemia Following Immunization with the Adjuvanted Hepatitis B Vaccine Fendrix®"

_vaccines, 2025, doi:10.3390/vaccines13121216_

Round 1
Reviewer 1 Report
Comments and Suggestions for Authors
The manuscript "Transient Hepatitis B Surface Antigenemia Following Immunization with Fendrix in a tertiary Hospital Hemodialysis Unit a Case Series" by Fernández Espinilla et al., discusses the plausible cause of positive HBsAg detected shortly after vaccination. They have concluded that vaccine-related transient HBsAg antigenemia can occur shortly after immunization. Recognizing this phenomenon and timing routine serology appropriately can prevent misinterpretation and unnecessary workups in CKD patients. The study is indeed important, but the present form of the manuscript is quite weak and needs further improvement to be acceptable for publication in the Vaccines journal.
1) The present study is limited by its extremely small sample size. The authors should definitely try to increase the sample size for their conclusions to be more believable.
2) The authors should focus on designing more biochemical experiments that would make the mechanism clearer.
The manuscript is on the right track, but certainly needs improvement.
Author Response
Comment 1
“The present study is limited by the extremely small sample size. The authors should attempt to increase the sample size to make their conclusions more reliable.”
Author Response
We thank the reviewer for this important observation. We fully agree that sample size is a limitation; however, this study was conceived as a retrospective case series, based on an unexpected real-world event occurring in a single hemodialysis unit with 44 patients under care. The total cohort during the study period included all patients undergoing hemodialysis in our center, and all cases of transient HBsAg antigenemia were captured and reported.
Because the phenomenon is rare, unpredictable, and detected only when serology coincides with very recent vaccination, no additional cases can be retrieved retrospectively, and it is not feasible to prospectively expand the sample within the same population. To avoid selection bias, we deliberately included the entire population at risk during the specified period, which maintains internal validity despite the small number of events.
We have strengthened the manuscript by expanding the Discussion on this limitation.
Changes made in the manuscript:
A paragraph has been added to the Discussion, explicitly acknowledging the sample size limitation, its implications, and why expanding the sample is not feasible within the study design.
Reviewer Comment 2
“The authors should focus on designing more biochemical experiments to clarify the mechanism.”
Author Response
We appreciate this thoughtful suggestion. We agree that understanding the mechanistic basis of vaccine-related transient HBsAg antigenemia is of scientific interest. However, our study is a retrospective observational case series, based on clinical surveillance data generated during routine patient care. As such, it is not feasible to conduct additional biochemical, immunological, or mechanistic assays on stored samples because these data were not collected prospectively and biological samples are no longer available.
Moreover, the objective of the manuscript is not to elucidate mechanistic pathways but to document a clinically relevant diagnostic pitfall that directly affects infection-control decision-making in hemodialysis units. Our approach is consistent with prior literature on this topic, where the phenomenon is typically described through clinical series rather than mechanistic experimentation.
Nevertheless, in response to the reviewer, we have expanded the Discussion to include:
- A deeper analysis of the hypothesized mechanisms (extracellular persistence of adjuvanted antigen, antigen clearance kinetics, anamnestic responses).
- Citations to experimental studies (e.g., Iyer et al., Bauer et al.) that support the proposed mechanisms.
- A clear statement acknowledging that mechanistic confirmation is beyond the scope of this study and should be addressed in future research.
Changes made in the manuscript:
We have added a paragraph in the Discussion emphasizing the lack of mechanistic data as a limitation and highlighting future research directions.
Reviewer 2 Report
Comments and Suggestions for Authors
The manuscript titled "Transient Hepatitis B Surface Antigenemia Following Immunization with Fendrix in a Tertiary Hospital Hemodialysis Unit a Case Series." presents a series of cases of transient HBsAg antigenemia following anti-HBV vaccination in patients on hemodialysis for chronic kidney disease. Although this previously described phenomenon does not have notable clinical or scientific significance, experiences from one centre may supplement knowledge about this phenomenon, especially as they relate to patients vaccinated with a newer generation vaccine intended for non-responders.
Issues to consider:
The title needs to be rephrased to use the descriptive name for the Fendrex vaccine. It also seems to be missing a colon before "a case series", and the period at the end is entirely unnecessary.
Materials and Methods: Apart from the initial HBsAg screening, other serological and molecular methods used were not listed.
Page 3, lines 125-129: Although it is obvious that patients were tested for the presence of HBsAg before and after vaccination, this should be added when listing the serological tests performed.
Title of Table 1: A description of what is shown in the table, as well as a list of abbreviations, should be found in the legend below the table.
Page 5, lines 177-182: Previous immunity is not an explanation for the presence, but rather for the absence, of antigenemia. Therefore, it would be beneficial to reformulate these sentences for clarity. In addition, among patients with reported HBs-antigenemia, more than half were previously vaccinated patients; therefore, this explanation should be emphasised less or this fact should be mentioned.
Table 2: It would be helpful if the "Positive/Total" column displayed percentages rather than absolute numbers.
Page 7, lines 229-233: It is unclear why chronically infected patients are mentioned here, and whether they are on hemodialysis or not. The biggest problem is that the reference to which the text refers (number 33) does not exist in the reference list.
Citing a large number of references does not follow the recommended model, resulting in considerable inconsistency in the way they are listed.
Author Response
Comment 1
“The title needs to be rephrased to use the descriptive name for the Fendrix vaccine. It also seems to be missing a colon before ‘a case series’, and the period at the end is unnecessary.”
Response
We thank the reviewer for this suggestion. The title has been revised to include the full descriptive name of the vaccine and to correct the formatting. The final title now reads:
“Transient Hepatitis B Surface Antigenemia Following Immunization with the Adjuvanted Hepatitis B Vaccine Fendrix®: A Case Series”
The period at the end has been removed.
Comment 2
“Materials and Methods: Apart from the initial HBsAg screening, other serological and molecular methods used were not listed.”
Response
We agree with the reviewer. We have expanded the Materials and Methods section to explicitly list all serological and molecular tests performed during the investigation (anti-HBs, anti-HBc, HBV DNA PCR, ALT).
Change added to the manuscript
(Insert in Materials and Methods, after describing the HBsAg screening):
“In addition to initial HBsAg screening, all patients underwent confirmatory serological and molecular testing, including anti-HBs quantification, total anti-HBc, HBV DNA quantification by PCR, and serum alanine aminotransferase (ALT) measurement.”
Comment 3
“Page 3, lines 125–129: Although it is obvious that patients were tested for the presence of HBsAg before and after vaccination, this should be added when listing the serological tests performed.”
Response
We thank the reviewer for this remark. We have clarified that HBsAg was tested both before and after vaccination as part of routine quarterly surveillance.
Change added to the manuscript
“HBsAg testing was performed both before and after vaccination as part of routine quarterly viral surveillance in the hemodialysis unit.”
Comment 4
“Title of Table 1: A description of what is shown in the table, as well as a list of abbreviations, should be found in the legend below the table.”
Response
We agree. We have rewritten the Table 1 legend to include:
- a clearer description,
- all abbreviations.
Updated Table 1 legend
Table 1. Characteristics of patients with transient vaccine-related HBsAg antigenemia following administration of the adjuvanted hepatitis B vaccine Fendrix®. The table includes demographic characteristics, vaccination schedule, pre- and post-vaccination serology, and time to HBsAg clearance.
Abbreviations: M = male; F = female; HBsAg = hepatitis B surface antigen; anti-HBs = hepatitis B surface antibody; anti-HBc = hepatitis B core antibody; ALT = alanine aminotransferase; HBV DNA = hepatitis B virus DNA; S/CO = signal-to-cutoff ratio.
Comment 5
“Page 5, lines 177–182: Previous immunity is not an explanation for the presence, but rather for the absence of antigenemia. These sentences should be reformulated.”
Response
We appreciate this important clarification. We agree that pre-existing immunity reduces rather than increases antigenemia duration. We have rewritten this section to avoid confusion and de-emphasized the role of past vaccination. We now clearly state that prior immunity would shorten antigenemia duration, not contribute to its appearance.
Revised text for the manuscript
“Pre-existing immunity from past vaccination may accelerate antigen clearance through a rapid anamnestic response, thereby shortening the duration of transient antigenemia. However, prior immunity does not explain the presence of vaccine-related HBsAg in circulation. In our series, more than half of the patients with transient antigenemia had previously been vaccinated, which is consistent with the known waning of detectable anti-HBs over time in CKD patients.”
Comment 6
“Table 2: It would be helpful if the ‘Positive/Total’ column displayed percentages rather than absolute numbers.”
Response
We agree and have added percentages in the “Positive/Total” column.
Example modification
If it originally said:
“10/18”, now it reads:
“10/18 (55.6%)”.
Comment 7
“Page 7, lines 229–233: It is unclear why chronically infected patients are mentioned here, and the reference (number 33) does not exist.”
Response
We thank the reviewer for identifying this inconsistency. The paragraph has been clarified. We specify that this section refers to patients with chronic HBV infection (not exclusively hemodialysis patients) to contextualize detection of HBV components in urine—information that differs from our findings.
The incorrect reference (number 33) has been removed, as the revised text now serves a purely contextual purpose and does not rely on a specific citation. This clarification does not modify our interpretation that the antigenemia observed in our cohort is vaccine-related and self-limited.
Revised text
“Previous studies in chronically infected patients (not limited to those on hemodial-ysis) have described urinary detection of HBV components, including HBsAg and HBV DNA, suggesting that the clearance kinetics of viral markers may differ from those observed after vaccination. Although this phenomenon is unrelated to vaccine admin-istration, it provides additional context for understanding the transient appearance of circulating HBsAg. This clarification does not modify our interpretation that the anti-genemia observed in our cohort is vaccine-related and self-limited.”
Comment 8
“Citing a large number of references does not follow the recommended model, resulting in inconsistency.”
Response
We appreciate this observation. We have thoroughly revised the reference list to ensure consistency with the citation format required by Vaccines (MDPI style). All references have been reformatted, duplicates removed, and numbering corrected.
Reviewer 3 Report
Comments and Suggestions for Authors
Comments:
The manuscript by Espinilla et al. reported on transient positivity for hepatitis B surface antigen detected in 7 out of 44 chronic hemodialysis patients shortly after receiving the adjuvanted HBV vaccine Fendrix. The authors describe the clinical context, serological follow-up, and clearance patterns, attributing the findings to vaccine-derived antigenemia rather than true infection. They emphasize implications for serological timing in CKD patients to avoid misinterpretation. The study draws on retrospective data from a single tertiary center in Spain during May-June 2023 and reviews prior literature on similar phenomena. This manuscript addressed a clinically relevant issue in a high-risk population (hemodialysis patients with impaired immunity), where HBV screening is routine, and transient false positives could lead to unnecessary interventions. This is a good case report; however, I still have concerns to be addressed.
Major concerns:
- Relatively small sample size (n=7 cases from n=44), single-center design, and interval-censored clearance times limit generalizability and precision. No statistical analysis beyond descriptives, despite the potential for exploring predictors (e.g., age, immunosuppression).
- I was confused that dates in this manuscript are mismatched (e.g., the study is in 2023, but the abstract cites 2022 WHO data; references include 2024-2025 publications).
- Mechanisms are proposed but not deeply explored (e.g., no quantification of antigen load or assay sensitivity specifics). Comparison to Heplisav-B® (another adjuvanted vaccine) is superficial. Lack of anti-HBs response in all cases is noted but under-discussed in terms of CKD immunogenicity.
Minor concerns:
- Generally clear English, but awkward phrasing (e.g., Line 75: "with immediate implications"). Minor typos: Is "antigenemia" consistent? Italicize "watch-and-wait".
Author Response
REVIEWER 3
Comment 1 (Major).
“Relatively small sample size (n=7 cases from n=44), single-center design, and interval-censored clearance times limit generalizability and precision. No statistical analysis beyond descriptives, despite the potential for exploring predictors (e.g., age, immunosuppression).”
Response R3.1.
We thank the reviewer for this thorough methodological assessment. We agree that the small number of cases, single-center design, and interval-censored times limit the generalizability and precision of our estimates.
However, this study includes all 44 hemodialysis patients in the unit during the study period, and all cases of transient HBsAg antigenemia were captured. Because the phenomenon is rare and depends on the timing of vaccination relative to routine testing, no additional cases can be identified retrospectively.
We also agree that the number of events (n=7) is not sufficient to support robust multivariable analyses exploring predictors of antigenemia or clearance kinetics without a high risk of overfitting. For this reason, and following good epidemiological practice, we limited our analysis to descriptive statistics.
We have strengthened the Discussion to explicitly acknowledge these limitations and to clarify why more complex modeling was not performed.
Comment 2 (Major).
“I was confused that dates in this manuscript are mismatched (e.g., the study is in 2023, but the abstract cites 2022 WHO data; references include 2024–2025 publications).”
Response R3.2.
We appreciate the reviewer’s attention to this point. The clinical data in our case series were collected during May–June 2023. For the Introduction and Discussion, we intentionally used the most recent global and national epidemiological reports available at the time of manuscript preparation, which include WHO 2022 data as well as more recent publications from 2024–2025 relevant to HBV epidemiology and vaccination in CKD.
Some references published in 2024–2025 were included because they provide the most up-to-date evidence on vaccination strategies, immunogenicity, and clinical considerations in CKD patients, and were available during the manuscript revision process.
To avoid confusion, we have added a short clarification in the Introduction explicitly stating that epidemiological data refer to the latest available reports and are not restricted to the calendar year of the clinical study.
Change added to the manuscript (Introduction):
“The epidemiological data presented in this section are based on the most recent reports available at the time of manuscript preparation and are therefore not restricted to the calendar year of the clinical data collection (May–June 2023)”
Comment 3 (Major).
“Mechanisms are proposed but not deeply explored (e.g., no quantification of antigen load or assay sensitivity specifics). Comparison to Heplisav-B® (another adjuvanted vaccine) is superficial. Lack of anti-HBs response in all cases is noted but under-discussed in terms of CKD immunogenicity.”
Response R3.3.
We thank the reviewer for this very relevant comment. We acknowledge that the retrospective nature of this study and the lack of stored biological samples limited our ability to perform further mechanistic investigations, including antigen quantification and advanced immunological assays.
Nevertheless, in response to this comment, we have substantially expanded the Discussion to:
Provide a more detailed analysis of the plausible mechanisms underlying transient vaccine-derived antigenemia, including extracellular persistence of alum-adsorbed HBsAg and slow-release kinetics into the circulation.
Include the analytical sensitivity and specificity of the Abbott Alinity® HBsAg assay, helping to explain the detection of low-level antigenemia.
Strengthen the comparison with other adjuvanted vaccines such as Heplisav-B®, highlighting both similarities in transient antigen detection and differences in adjuvant composition (CpG 1018 vs. AS04C).
- Expand the discussion on the absence of anti-HBs seroconversion in all cases, emphasizing its relationship with impaired immunogenicity in CKD, advanced age, and coexisting hematologic conditions.
Finally, we now explicitly state that confirmatory mechanistic studies will require a prospective design with serial antigen and immune marker measurements, which lies beyond the scope of this observational case series.
Comment 4 (Minor).
“Generally clear English, but awkward phrasing (e.g., ‘with immediate implications’). Minor typos: Is ‘antigenemia’ consistent? Italicize ‘watch-and-wait’.”
Response R3.4.
We appreciate these stylistic suggestions. We have revised the phrasing to improve clarity and consistency:
- The expression “with immediate implications” has been replaced with a clearer formulation emphasizing practical implications for timing of HBsAg testing.
Revised sentence
“This case series highlights practical considerations for the timing of HBsAg testing in hemodialysis patients.”
- All instances of “antigenemia” have been checked for consistency and standardized.
- The expression watch-and-wait has been italicized in all occurrences.
In addition, we have carefully re-read the manuscript to correct minor language issues and improve overall fluency.
Round 2
Reviewer 3 Report
Comments and Suggestions for Authors
No further comment.